# BREEDS: BENCHMARKS FOR SUBPOPULATION SHIFT

**Shibani Santurkar**[*]
MIT
shibani@mit.edu

**Dimitris Tsipras**[*]
MIT
tsipras@mit.edu

**Aleksander Mądry**
MIT
madry@mit.edu

## ABSTRACT

We develop a methodology for assessing the robustness of models to subpopulation shift—specifically, their ability to generalize to novel data subpopulations that were not observed during training. Our approach leverages the class structure underlying existing datasets to control the data subpopulations that comprise the training and test distributions. This enables us to synthesize realistic distribution shifts whose sources can be precisely controlled and characterized, within existing large-scale datasets. Applying this methodology to the ImageNet dataset, we create a suite of subpopulation shift benchmarks of varying granularity. We then validate that the corresponding shifts are tractable by obtaining human baselines. Finally, we utilize these benchmarks to measure the sensitivity of standard model architectures as well as the effectiveness of existing train-time robustness interventions. [1]

## 1 INTRODUCTION

Robustness to distribution shift has been the focus of a long line of work in machine learning (Schlimmer & Granger, 1986; Widmer & Kubat, 1993; Kelly et al., 1999; Shimodaira, 2000; Sugiyama et al., 2007; Quionero-Candela et al., 2009; Moreno-Torres et al., 2012; Sugiyama & Kawanabe, 2012). At a high-level, the goal is to ensure that models perform well not only on unseen samples from the datasets they are trained on, but also on the diverse set of inputs they are likely to encounter in the real world. However, building benchmarks for evaluating such robustness is challenging—it requires modeling realistic data variations in a way that is well-defined, controllable, and easy to simulate.

Prior work in this context has focused on building benchmarks that capture distribution shifts caused by natural or adversarial input corruptions (Szegedy et al., 2014; Fawzi & Frossard, 2015; Fawzi et al., 2016; Engstrom et al., 2019b; Ford et al., 2019; Hendrycks & Dietterich, 2019; Kang et al., 2019), differences in data sources (Saenko et al., 2010; Torralba & Efros, 2011; Khosla et al., 2012; Tommasi & Tuytelaars, 2014; Recht et al., 2019), and changes in the frequencies of data subpopulations (Oren et al., 2019; Sagawa et al., 2020). While each of these approaches captures a different source of real-world distribution shift, we cannot expect any single benchmark to be comprehensive. Thus, to obtain a holistic understanding of model robustness, we need to keep expanding our testbed to encompass more natural modes of variation. In this work, we take another step in that direction by studying the following question:

*How well do models generalize to data subpopulations they have not seen during training?*

The notion of *subpopulation shift* this question refers to is quite pervasive. After all, our training datasets will inevitably fail to perfectly capture the diversity of the real word. Hence, during deployment, our models are bound to encounter unseen subpopulations—for instance, unexpected weather conditions in the self-driving car context or different diagnostic setups in medical applications.

### OUR CONTRIBUTIONS

The goal of our work is to create large-scale subpopulation shift benchmarks wherein the data subpopulations present during model training and evaluation differ. These benchmarks aim to

---

[*]Equal contribution.
[1]Code and data available at `https://github.com/MadryLab/BREEDS-Benchmarks`.

assess how effectively models generalize beyond the limited diversity of their training datasets—e.g., whether models can recognize Dalmatians as "dogs" even when their training data for "dogs" comprises only Poodles and Terriers. We show how one can simulate such shifts, fairly naturally, *within* existing datasets, hence eliminating the need for (and the potential biases introduced by) crafting synthetic transformations or collecting additional data.

**BREEDS benchmarks.** The crux of our approach is to leverage existing dataset labels and use them to identify *superclasses*—i.e., groups of semantically similar classes. This allows us to construct classification tasks over such superclasses, and repurpose the original dataset classes to be the subpopulations of interest. This, in turn, enables us to induce a subpopulation shift by directly making the subpopulations present in the training and test distributions disjoint. By applying this methodology to the ImageNet dataset (Deng et al., 2009), we create a suite of subpopulation shift benchmarks of varying difficulty. This involves modifying the existing ImageNet class hierarchy—WordNet (Miller, 1995)—to ensure that superclasses comprise visually coherent subpopulations. We conduct human studies to validate that the resulting benchmarks capture meaningful subpopulation shifts.

**Model robustness to subpopulation shift.** In order to demonstrate the utility of our benchmarks, we employ them to evaluate the robustness of standard models to subpopulation shift. In general, we find that model performance drops significantly on the shifted distribution—even when this shift does not significantly affect humans. Still, models that are more accurate on the original distribution tend to also be more robust to these subpopulation shifts. Moreover, adapting models to the shifted domain, by retraining their last layer on this domain, only partially recovers the original model performance.

**Impact of robustness interventions.** Finally, we examine whether various train-time interventions, designed to decrease model sensitivity to synthetic data corruptions (e.g., $\ell_2$-bounded perturbations) make models more robust to subpopulation shift. We find that many of these methods offer small, yet non-trivial, improvements along this axis—at times, at the expense of performance on the original distribution. Often, these improvements become more pronounced after retraining the last layer of the model on the shifted distribution. Nevertheless, the increase in model robustness to subpopulation shifts due to these interventions is much smaller than what is observed for other families of input variations such as data corruptions (Hendrycks & Dietterich, 2019; Ford et al., 2019; Kang et al., 2019; Taori et al., 2020). This indicates that handling subpopulation shifts, such as those present in the BREEDS benchmarks, might require a different set of robustness tools.

## 2 DESIGNING BENCHMARKS FOR DISTRIBUTION SHIFT

When constructing distribution shift benchmarks, the key design choice lies in specifying the *target distribution* to be used during model evaluation. This distribution is meant to be a realistic variation of the *source distribution*, that was used for training. Typically, studies focus on variations due to:

- *Data corruptions*: The target distribution is obtained by modifying inputs from the source distribution via a family of transformations that mimic real-world corruptions, as in Fawzi & Frossard (2015); Fawzi et al. (2016); Engstrom et al. (2019b); Hendrycks & Dietterich (2019); Ford et al. (2019); Kang et al. (2019); Shankar et al. (2019).

- *Differences in data sources*: Here, the target distribution is an independent dataset for the same task (Saenko et al., 2010; Torralba & Efros, 2011; Tommasi & Tuytelaars, 2014; Recht et al., 2019)—e.g., collected at a different geographic location (Beery et al., 2018), time frame (Kumar et al., 2020) or user population (Caldas et al., 2018). For instance, this could involve using PASCAL VOC (Everingham et al., 2010) to evaluate Caltech101-trained classifiers (Fei-Fei et al., 2006). The goal is to test whether models are overly reliant on the idiosyncrasies of their training datasets (Ponce et al., 2006; Torralba & Efros, 2011).

- *Subpopulation representation*: The source and target distributions differ in terms of how well-represented each subpopulation is. Work in this area typically studies whether models perform *equally well* across all subpopulations from the perspective of reliability (Meinshausen et al., 2015; Hu et al., 2018; Duchi & Namkoong, 2018; Caldas et al., 2018; Oren et al., 2019; Sagawa et al., 2020) or algorithmic fairness (Dwork et al., 2012; Kleinberg et al., 2017; Jurgens et al., 2017; Buolamwini & Gebru, 2018; Hashimoto et al., 2018).

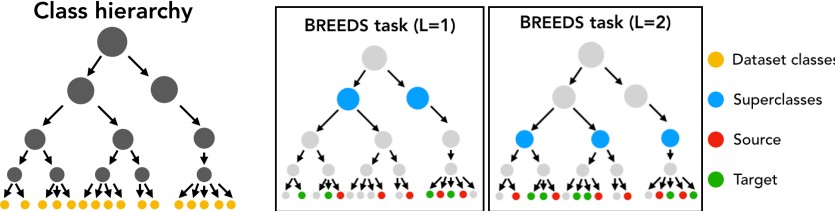

Figure 1: Illustration of our pipeline to create subpopulation shift benchmarks. Given a dataset, we define superclasses based on the semantic hierarchy of dataset classes. This allows us to treat the dataset labels as subpopulation annotations. Then, we construct a BREEDS task of specified granularity (i.e., depth in the hierarchy) by posing the classification task in terms of superclasses at that depth and then partitioning their respective subpopulations into the source and target domains.

These variations simulate realistic ways in which the data encountered during deployment can deviate from training conditions. However, each of the aforementioned benchmarks capture only one facet of real-world distribution shifts. It is not clear a priori that robustness to any subset of these variations will necessarily translate to robustness with respect to the rest. Thus, to effectively assess and improve model robustness, we require a varied suite of distribution shift benchmarks.

## 3  THE BREEDS METHODOLOGY

In this work, we focus on modeling a pertinent, yet less studied, form of subpopulation shift: one wherein the target distribution (used for testing) contains subpopulations that are *entirely* absent from the source distribution that the model was trained on. To simulate such shifts, we need to precisely control the data subpopulations present in the source and target data distributions. Our procedure for doing this comprises two stages that are outlined below—see Figure 1 for an illustration and Appendix A.2 for pseudocode.

**Devising subpopulation structure.**  Typical datasets do not contain annotations for individual subpopulations. Since collecting such annotations would be challenging, we take an alternative approach: we bootstrap the existing dataset labels to simulate subpopulations. That is, we group semantically similar classes into broader superclasses which, in turn, allows us to re-purpose existing class labels as the desired subpopulation annotations. Moreover, we can group classes in a hierarchical manner, obtaining superclasses of different specificity. As we will see in Section 4, such class hierarchies are already present in large-scale benchmarks (Deng et al., 2009; Kuznetsova et al., 2018).

**Simulating subpopulation shifts.**  Given a set of superclasses, we can define a classification task over them: the inputs of each superclass correspond to pooling together the inputs of its subclasses (i.e., the original dataset classes). Within this setup, we can simulate subpopulation shift in a relatively straightforward manner. Specifically, for each superclass, we split its subclasses into two *random* and *disjoint* sets, and assign one of them to the source and the other to the target domain. Then, we can evaluate model robustness under subpopulation shift by simply training on the source domain and testing on the target domain. Note that the classification task remains identical between domains—both domains contain the same (super)classes but the subpopulations that comprise each (super)class differ. [2] Intuitively, this corresponds to using different dog breeds to represent the class "dog" during training and testing—hence the name of our toolkit.

This methodology is quite general and can be applied to a variety of setting to simulate realistic distribution shifts. Moreover, it has a number of additional benefits:

- **Flexibility:** Different semantic groupings of a fixed set of classes lead to BREEDS tasks of varying granularity. For instance, by only grouping together classes that are quite similar

---

[2]Note that this approach can be extended to simulate milder subpopulation shifts where the source and target distributions overlap but the relative subpopulation frequencies vary, similar to the setting of Oren et al. (2019).

one can reduce the severity of the subpopulation shift. Alternatively, one can consider broad superclasses, each having multiple subclasses, resulting in a more challenging benchmark.

- **Precise characterization:** The exact subpopulation shift between the source and target domains is known. Since both domains are constructed from the same dataset, the impact of any external factors (e.g., differences in data collection pipelines) is minimized. Note that such external factors can significantly impact the difficulty of the task (Ponce et al., 2006; Torralba & Efros, 2011; Tsipras et al., 2020). In fact, minimizing these effects and ensuring that the shift between the source and target domain is caused solely by the intended input variations is one of the major challenges in building distribution shift benchmarks. For instance, recent work (Engstrom et al., 2020) demonstrates that statistical biases during data collection can significantly skew the intended target distribution.

- **Symmetry:** Since subpopulations are split into the source and test domains randomly, we expect the resulting tasks to have comparable difficulty.

- **Reuse of existing datasets:** No additional data collection or annotation is required other than choosing the class grouping. This approach can thus be used to also re-purpose other existing large-scale datasets—even beyond image recognition—with minimal effort.

## 4 SIMULATING SUBPOPULATION SHIFTS WITHIN IMAGENET

We now describe how our methodology can be applied to ImageNet (Deng et al., 2009)—specifically, the ILSVRC2012 subset (Russakovsky et al., 2015)—to create a suite of BREEDS benchmarks. ImageNet contains a large number of classes, making it particularly well-suited for our purpose.

### 4.1 UTILIZING THE IMAGENET CLASS HIERARCHY

Recall that creating BREEDS tasks requires grouping together similar classes. For ImageNet, such a semantic grouping already exists—ImageNet classes are a part of the WordNet hierarchy (Miller, 1995). However, WordNet is not a hierarchy of objects but rather one of word meanings. Thus, intermediate hierarchy nodes are not always well-suited for object recognition due to:

- **Abstract groupings:** WordNet nodes often correspond to abstract concepts, e.g., related to the functionality of an object. Children of such nodes might thus share little visual similarity—e.g., "umbrella" and "roof" are visually different, despite both being "coverings."

- **Non-uniform categorization:** The granularity of object categorization is vastly different across the WordNet hierarchy—e.g., the subtree rooted at "dog" is 25-times larger than the one rooted at "cat." Hence, the depth of a node in this hierarchy does not always reflect the specificity of the corresponding object category.

- **Lack of tree structure:** Nodes in WordNet can have multiple parents and thus the resulting classification task would contain overlapping classes, making it inherently ambiguous.

Due to these issues, we cannot directly use WordNet to identify superclasses that correspond to a well-calibrated classification task. To illustrate this, we present some of the superclasses that Huh et al. (2016) constructed by applying clustering algorithms directly to the WordNet hierarchy in Appendix Table 7. Even putting the issue of overlapping classes aside, a BREEDS task based on these superclasses would induce a very skewed subpopulation shift across classes—e.g., varying the types of "bread" is very different that doing the same for different "mammal" species.

To better align the WordNet hierarchy with the task of object recognition in general, and BREEDS benchmarks in particular, we manually modify it according to the following two principles: (i) nodes should be grouped together based on their visual characteristics rather than abstract relationships like functionality, and (ii) nodes of similar specificity should be at the same distance from the root, irrespective of how detailed their categorization within WordNet is. Details of this procedure along with the resulting hierarchy are presented in Appendix A.4.

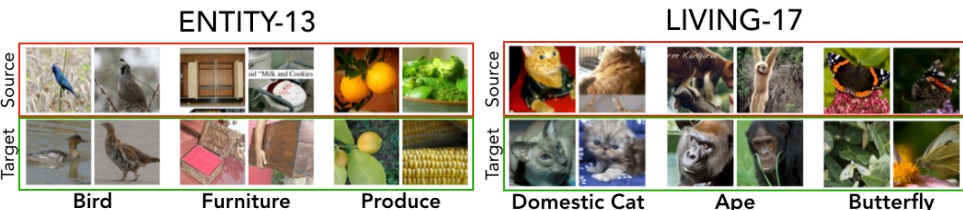

Figure 3: Sample images from random object categories for the ENTITY-13 and LIVING-17 tasks. For each task, the top and bottom row correspond to the source and target distributions respectively.

## 4.2 CREATING BREEDS TASKS

Once the modified version of the WordNet hierarchy is in place, BREEDS tasks can be created in an automated manner. Specifically, we first choose the desired granularity of the task by specifying the distance from the root ("entity") and retrieving all superclasses at that distance in a top-down manner. Each resulting superclass corresponds to a subtree of our hierarchy, with ImageNet classes as its leaves. Note that these superclasses are roughly of the same specificity, due to our hierarchy restructuring process. Then, we randomly sample a fixed number of subclasses for each superclass to produce a balanced dataset (omitting superclasses with an insufficient number of subclasses). Finally, as described in Section 3, we randomly split these subclasses into the source and target domain. [3]

For our analysis, we create four tasks (cf. Table 2) based on different levels/parts of the hierarchy. To illustrate what the corresponding subpopulation shifts look like, we present (random) image samples for a subset of the tasks in Figure 3. Note that while we focus on the tasks in Table 2 in our study, our methodology readily enables us to create other variants of these tasks in an automated manner.

| Name | Subtree | Level | Subpopulations | Examples |
|------|---------|-------|----------------|----------|
| ENTITY-13 | "entity" (root) | 3 | 20 | "mammal", "appliance" |
| ENTITY-30 | "entity" (root) | 4 | 8 | "fruit", "carnivore" |
| LIVING-17 | "living thing" | 5 | 4 | "ape", "bear" |
| NON-LIVING-26 | "non-living thing" | 5 | 4 | "fence", "ball" |

Table 2: BREEDS benchmarks constructed using ImageNet. Here, "level" indicates the depth of the superclasses in the class hierarchy (task granularity), and the number of "subpopulations" (per superclass) is fixed to create balanced datasets. We also construct specialized tasks by focusing on subtrees in the hierarchy, e.g., only living (LIVING-17) or non-living (NON-LIVING-26) objects. Datasets naming reflects the root of the subtree and the number of superclasses they contain.

**BREEDS benchmarks beyond ImageNet.** It is worth nothing that the methodology we described is not restricted to ImageNet and can be readily applied to other datasets as well. The only requirement is that we have access to a semantic grouping of the dataset classes, which is the case for many popular vision datasets—e.g., CIFAR-100 (Krizhevsky, 2009), Pascal-VOC (Everingham et al., 2010), OpenImages (Kuznetsova et al., 2018), COCO-Stuff (Caesar et al., 2018). Moreover, even when a class hierarchy is entirely absent, the needed semantic class grouping can be manually constructed with relatively little effort (proportional to the number of classes, not the number of datapoints).

More broadly, the methodology of utilizing existing dataset annotations to construct data subpopulations goes beyond image classification tasks. In particular, by splitting inputs into a source and target domain based on some attribute, we can measure how well models generalize along this axis. Examples would include grouping by brand in Amazon reviews (McAuley et al., 2015), by location in Berkeley DeepDrive (Yu et al., 2020), and by facial attributes in CelebA (Liu et al., 2015).

---

[3]We also consider more benign or adversarial subpopulation splits for these tasks in Appendix C.2.1.

## 4.3 CALIBRATING BREEDS BENCHMARKS VIA HUMAN STUDIES

For a distribution shift benchmark to be meaningful, it is essential that the source and target domains capture the same high-level task—otherwise generalizing from one domain to the other would be impossible. To ensure that this is the case for the BREEDS task, we assess how significant the resulting distribution shifts are for human annotators (crowd-sourced via MTurk).

**Annotator task.** To obtain meaningful performance estimates, it is crucial that annotators perform the task based only *on the visual content of the images*, without leveraging prior knowledge. To achieve this, we design the following annotation task. First, annotators are shown images from the source domain, grouped by superclass, without being aware of the superclass name (i.e., the grouping it corresponds to). Then, they are presented with images from the target domain and are asked to assign each of them to one of the groups. For simplicity, we present two random superclasses at a time, effectively simulating binary classification. Annotator accuracy can be measured directly as the fraction of images that they assign to the superclass to which they belong. We perform this experiment for each of the BREEDS tasks constructed in Section 4.2. For comparison, we repeat this experiment without subpopulation shift (test images are sampled from the source domain) and for the superclasses constructed by Huh et al. (2016) using the WordNet hierarchy directly (cf. Appendix A.6).

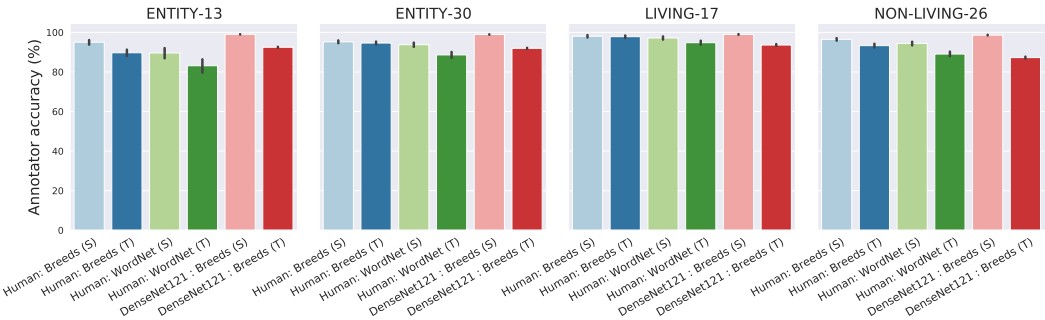

Figure 4: Human performance on (binary) BREEDS tasks. Annotators are provided with labeled images from the source distribution for a *pair of (undisclosed) superclasses*, and asked to classify samples from the target domain ('T') into one of the two groups. As a baseline we also measure annotator performance without subpopulation shift (i.e., on test images from the source domain, 'S') and tasks created via the WordNet hierarchy (cf. Appendix A.6). We observe that annotators are fairly robust to subpopulation shift. Further, they consistently perform better on BREEDS task compared to those based on WordNet directly—indicating that our modified class hierarchy is indeed better calibrated for object recognition. (We discuss model performance in Section 5.)

**Human performance.** We find that, across all tasks, annotators perform well on unseen data from the source domain, as expected. More importantly, annotators also appear to be quite robust to subpopulation shift, experiencing only a small accuracy drop between the source and target domains (cf. Figure 5). This indicates that the source and target domains are indeed perceptually similar for humans, making these benchmarks suitable for studying model robustness. Finally, across all benchmarks, annotators perform better on BREEDS tasks, compared to their WordNet equivalents—even on source domain samples. This indicates that our modified class hierarchy is indeed better aligned with the underlying visual recognition task.

## 5 EVALUATING MODEL PERFORMANCE UNDER SUBPOPULATION SHIFT

We can now use our suite of BREEDS tasks as a testbed for assessing model robustness to subpopulation shift as well as gauging the effectiveness of various train-time robustness interventions. Specifics of the evaluation setup and additional experimental results are provided in Appendices A.7 and C.2.

## 5.1 STANDARD TRAINING

We start by evaluating the performance of various model architectures trained in the standard fashion: empirical risk minimization (ERM) on the source distribution (cf. Appendix A.7.1). While models perform well on unseen inputs from the domain they are trained on, i.e., they achieve high *source accuracy*, their accuracy considerably drops under subpopulation shift—more than 30% in most cases (cf. Figure 5). At the same time, models that are more *accurate* on the source domain also appear to be more *robust* to subpopulation shift. Specifically, the fraction of source accuracy that is preserved in the target domain typically increases with source accuracy. (If this were not the case, i.e., the model accuracy dropped by a constant fraction under distribution shift, the target accuracy would match the baseline in Figure 5.) This indicates that, improvements in source accuracy *do* correlate with models generalizing better to variations in testing conditions.

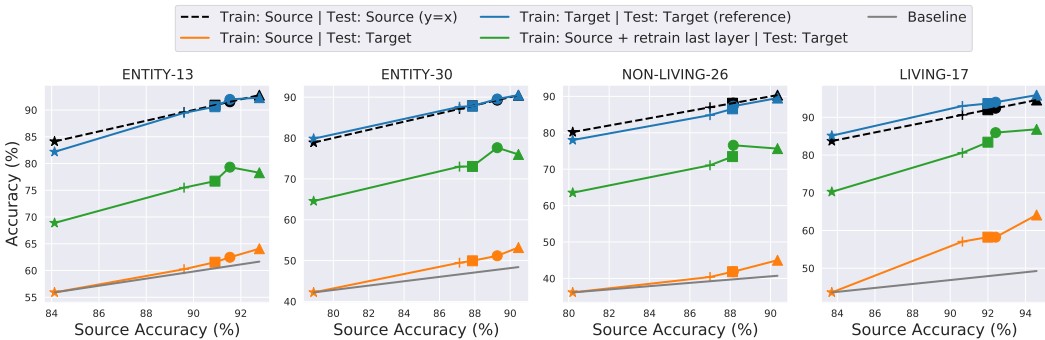

Figure 5: Robustness of standard models to subpopulation shifts. For each task, we plot the accuracy of various model architectures (denoted by different symbols) on the target domain as a function of their source accuracy. We find that model accuracy drops significantly between domains (*orange* vs. *dashed* line). Still, models that are more accurate on the source domain seem to also be more robust (the improvements exceed the baseline (*grey*) which would correspond to a constant accuracy drop relative to AlexNet). Moreover, the drop in model performance can be significantly (but not fully) reduced by retraining the final model layer with data from the target domain (*green*).

**Models vs. Humans.**   We compare the best performing model (DenseNet-121 in this case) to our previously obtained human baselines in Figure 4. To allow for a fair comparison, model accuracy is measured on pairwise superclass classification tasks (cf. Appendix A.7). We observe that models do exceedingly well on unseen samples from the source domain—significantly outperforming annotators under our task setup. At the same time, models also appear to be more brittle, performing worse than humans on the target domain of these binary BREEDS tasks, despite their higher source accuracy.

**Adapting models to the target domain.**   Finally, we focus on the intermediate data representations learned by these models, to assess how suitable they are for distinguishing classes in the target domain. To evaluate this, we retrain the last (fully-connected) layer of models trained on the source domain with data from the target domain. We find that the target accuracy of these models increases significantly after retraining, indicating that the learned representations indeed generalize to the target domain. However, we cannot match the accuracy of models trained directly (end-to-end) on the target domain—see Figure 5—demonstrating that there is significant room for improvement.

## 5.2 ROBUSTNESS INTERVENTIONS

We now turn our attention to existing methods for decreasing model sensitivity to specific synthetic perturbations. Our goal is to assess if these methods enhance model robustness to subpopulation shift too. Concretely, we consider the following families of interventions (cf. Appendix A.7.3 for details): (i) *adversarial training* to enhance robustness to worst-case $\ell_p$-bounded perturbations (in our case $\ell_2$) (Madry et al., 2018), (ii) *training on a stylized version of ImageNet* to encourage models to rely more on shape rather than texture (Geirhos et al., 2019), and (iii) *training with random noise* to make models robust to data corruptions (here, Gaussian and Erase noise (Zhong et al., 2020)).

Note that these methods can be viewed as ways of imposing a prior on the features that the model relies on (Heinze-Deml & Meinshausen, 2017; Geirhos et al., 2019; Engstrom et al., 2019a). That is, by rendering certain features ineffective during training (e.g., texture) they incentivize the model to utilize alternative ones (e.g., shape). Since different feature families may manifest differently in the target domain, such interventions could significantly impact model robustness to subpopulation shift.

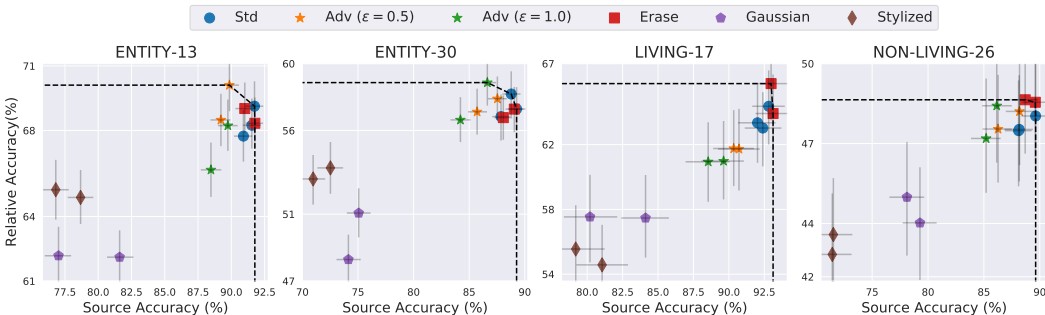

Figure 6: Effect of train-time interventions on model robustness to subpopulation shift. We measure model performance in terms of *relative accuracy*–i.e., the ratio between its target and source accuracies. This allows us to visualize the accuracy-robustness trade-off along with the corresponding Pareto frontier (*dashed*). (Also shown are 95% confidence intervals computed via bootstrapping.) We observe that some interventions do improve model robustness to subpopulation shift—specifically, erase noise and adversarial training—albeit by a small amount and often at the cost of source accuracy.

**Relative accuracy.**    To measure the impact of these interventions, we will focus on the models' *relative accuracy*—the ratio of target accuracy to source accuracy. This metric accounts for the fact that train-time interventions can impact model accuracy on the source domain itself. By measuring relative performance, we are able to compare different training methods on an equal footing.

We find that robustness interventions *do* have a small, yet non-trivial, impact on the robustness of a model to subpopulation shift—see Figure 6. Specifically, for the case of adversarial training and erase noise, models often retain a larger fraction of their accuracy on the target domain compared to standard training, hence lying on the Pareto frontier of a robustness-accuracy trade-off. In fact, for some of these interventions, the target accuracy is slightly higher than models obtained via standard training, even without adjusting for their lower source accuracy (raw accuracies are in Appendix C.2.2). Nonetheless, it is important to note that *none* of these methods offer significant subpopulation robustness—relative accuracy is not improved by more than a few percentage points.

**Adapting models to the target domain.**    The impact of these interventions is more pronounced if we consider the accuracy of models after their last layer is retrained on the target domain (cf. Appendix Figure 21). In particular, we find that for adversarially robust models, retraining significantly boosts accuracy on the target domain—e.g., for LIVING-17 it is almost comparable to the initial source accuracy. This suggests that the feature priors imposed by these interventions incentivize models to learn representations that generalize to other domains—in line with recent results of Utrera et al. (2020); Salman et al. (2020). Moreover, we observe that models trained on stylized inputs perform consistently worse, suggesting that texture might be an important feature for these tasks.

## 6    RELATED WORK

In Section 2, we surveyed prior work on distribution shift benchmarks. Here, we discuss further the benchmarks most closely related to ours and defer discussing additional related work to Appendix B.

Our benchmarks can be viewed as an instance of *domain generalization*. However, we focus on generalizing between different distributions of real-world images (photographs). This is in contrast to typical domain generalization benchmarks that focus on generalizing between different stylistic representations, e.g., from cartoons to drawings. Hence, the only comparable benchmark would be VLCS (Ghifary et al., 2015), which is however significantly smaller in scale and granularity than our

benchmarks. In a similar vein, datasets used in federated learning (Caldas et al., 2018) can be viewed as subpopulation shift benchmarks since the users present during training and testing might differ. However, to the best of our knowledge, there has been no large-scale vision benchmark in this setting.

Hendrycks & Dietterich (2019), in Appendix G, also (manually) construct a classification task over superclasses and use ImageNet classes outside of ILSVRC2012 (ImageNet-1k) to measure "subtype robustness". (Unfortunately, these classes are no longer publicly available (Yang et al., 2019).) Compared to their work, we use a general methodology to create a broader suite of benchmarks. Also, our analysis of architectures and robustness interventions is significantly more extensive.

## 7  CONCLUSION

In this work, we develop a methodology for constructing large-scale subpopulation shift benchmarks. The motivation behind our BREEDS benchmarks is to test if models can generalize beyond the limited diversity of their training datasets—specifically, to novel data subpopulations. A major advantage of our approach is its generality. It can be applied to any dataset with a meaningful class structure— including tasks beyond classification (e.g., object detection) and domains other than computer vision (e.g., natural language processing). Moreover, the subpopulation shifts are induced in a manner that is both controlled and natural, without altering inputs synthetically or requiring new data.

By applying this approach to the ImageNet dataset, we construct a suite of benchmarks of varying difficulty, that we then use to assess model robustness and the efficacy of various train-time interventions. Further, we obtain human baselines for these tasks to both put model performance in context and validate that the corresponding subpopulation shifts do not significantly affect humans.

Overall, our results indicate that existing models still have a long way to go before they can fully tackle BREEDS subpopulation shifts, even using current robustness interventions. We thus believe that our methodology provides a useful tool for studying and improving model robustness to distribution shift—an increasingly pertinent topic for real-world deployments of machine learning models.

## ACKNOWLEDGEMENTS

We thank Andrew Ilyas and Sam Park for helpful discussions.

Work supported in part by the NSF grants CCF-1553428, CNS-1815221, the Google PhD Fellowship, and the Microsoft Corporation. This material is based upon work supported by the Defense Advanced Research Projects Agency (DARPA) under Contract No. HR001120C0015.

Research was sponsored by the United States Air Force Research Laboratory and was accomplished under Cooperative Agreement Number FA8750-19-2-1000. The views and conclusions contained in this document are those of the authors and should not be interpreted as representing the official policies, either expressed or implied, of the United States Air Force or the U.S. Government. The U.S. Government is authorized to reproduce and distribute reprints for Government purposes notwithstanding any copyright notation herein.

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

# A  EXPERIMENTAL SETUP

## A.1  DATASET

We perform our analysis on the ILSVRC2012 dataset (Russakovsky et al., 2015). This dataset contains a thousand classes from the ImageNet dataset (Deng et al., 2009) with an independently collected validation set. The classes are part of the broader hierarchy, WordNet (Miller, 1995), through which words are organized based on their semantic meaning. We use this hierarchy as a starting point of our investigation but modify it as described in Appendix A.5.

For all the BREEDS superclass classification tasks, the train and validation sets are obtained by aggregating the train and validation sets of the descendant ImageNet classes (i.e., subpopulations). Specifically, for a given subpopulation, the training and test splits from the original ImageNet dataset are used as is.

## A.2  PIPELINE FORMALIZATION

Recall that our process for evaluating model robustness under subpopulation shift (cf. Section 3) is as follows. We present the pseudocode for this process in Algorithm 1.

1. Choose a level in the hierarchy and use it to define a set of superclasses by grouping the corresponding dataset classes together. Note that the original dataset classes form the subpopulations of the superclasses.
2. For every superclass, select a (random) set of subpopulations (i.e., classes in the original dataset) and use them to train the model to distinguish between superclasses (we call this the source domain).
3. For every superclass, use the remaining unseen subpopulations (i.e., classes in the original dataset) to test how well the model can distinguish between the superclasses (target domain).

---

**Algorithm 1** The BREEDS methodology. Evaluating the training method $train$ on level $L$ of the hierarchy $H$—restricted to the subtree under $root$—using $N_{sub}$ subpopulations per superclass.

---

**function** createDatasets($H$, $L$, $N_{sub}$, $root$):
    $source, target \leftarrow [], []$
    **for** $node \in H$ **do**
        **if** $node.depth = L$ **and** $root \in node.ancestors$ **and** $len(node.leaves) \geq N_{sub}$ **then**
            $y \leftarrow node.label$
            $subclasses \leftarrow$ random.choice($node.leaves$, $N_{sub}$)
            **for** $(i,\ c)\ \in enumerate(subclasses)$ **do**
                **if** $i \leq N_{sub}\,/\,2$ **then**
                    $domain \leftarrow source$
                **else**
                    $domain \leftarrow target$
                **for** $x \in c.inputs$ **do**
                    $domain$.append((x, y))
    **return** $(source,\ target)$

**function** evaluateMethod($train, H, L, N_{sub}, root$):
    $source, target \leftarrow$ createDatasets ($H, L, N_{sub}, root$)
    $model \leftarrow train(source)$
    $correct, total \leftarrow 0, 0$
    **for** $(x, y) \in target$ **do**
        $correct$ += $(model(x) = y)$
        $total$ += 1
    $targetAccuracy \leftarrow \frac{correct}{total}$
    **return** $targetAccuracy$

---

## A.3 WordNet issues

As discussed in Section 4, WordNet is a semantic rather than a visual hierarchy. That is, object classes are arranged based on their meaning rather than their visual appearance. Thus, using intermediate nodes for a visual object recognition task is not straightforward. To illustrate this, we examine a sample superclass grouping created by Huh et al. (2016) via automated bottom-up clustering in Table 7.

| Superclass | Random ImageNet classes |
|---|---|
| **instrumentality** | fire engine, basketball, electric fan, wok, thresher, horse cart, harvester, balloon, racket, can opener, carton, gong, unicycle, toilet seat, carousel, hard disc, cello, mousetrap, neck brace, barrel |
| **man-made structure** | beacon, yurt, picket fence, barbershop, fountain, steel arch bridge, library, cinema, stone wall, worm fence, palace, suspension bridge, planetarium, monastery, mountain tent, sliding door, dam, bakery, megalith, pedestal |
| **covering** | window shade, vestment, running shoe, diaper, sweatshirt, breastplate, shower curtain, shoji, miniskirt, knee pad, apron, pajama, military uniform, theater curtain, jersey, football helmet, book jacket, bow tie, suit, cloak |
| **commodity** | espresso maker, maillot, iron, bath towel, lab coat, bow tie, washer, jersey, mask, waffle iron, mortarboard, diaper, bolo tie, seat belt, cowboy hat, wig, knee pad, vacuum, microwave, abaya |
| **organism** | thunder snake, stingray, grasshopper, barracouta, Newfoundland, Mexican hairless, Welsh springer spaniel, bluetick, golden retriever, keeshond, African chameleon, jacamar, water snake, Staffordshire bullterrier, Old English sheepdog, pelican, sea lion, wire-haired fox terrier, flamingo, green mamba |
| **produce** | spaghetti squash, fig, cardoon, mashed potato, pineapple, zucchini, broccoli, cauliflower, butternut squash, custard apple, pomegranate, strawberry, Granny Smith, lemon, head cabbage, artichoke, cucumber, banana, bell pepper, acorn squash |

Table 7: Superclasses constructed by Huh et al. (2016) via bottom-up clustering of WordNet to obtain 36 superclasses—for brevity, we only show superclasses with at least 20 ImageNet classes each.

First, we can notice that these superclasses have vastly different granularities. For instance, "organism" contains the entire animal kingdom, hence being much broader than "produce". Moreover, "covering" is rather abstract class, and hence its subclasses often share little visual similarity (e.g., "window shade", "pajama"). Finally, due to the abstract nature of these superclasses, a large number of subclasses overlap—"covering" and "commodity" share 49 ImageNet descendants.

## A.4 Manual calibration

We manually modify the WordNet hierarchy according to the following two principles so as to make it better aligned for visual object recognition.

1. Nodes should be grouped together based on their visual characteristics, rather than abstract relationships like functionality—e.g., we eliminate nodes that do not convey visual information such as "covering".

2. Nodes of similar specificity should be at the same distance from the root, irrespective of how detailed their categorization within WordNet is—for instance, we placed "dog" at the same level as "cat" and "flower", even though the "dog" sub-tree in WordNet is much larger.

Finally, we removed a number of ImageNet classes that did not naturally fit into the hierarchy. Concretely, we modified the WordNet hierarchy by applying the following operations:

- *Collapse node*: Delete a node from the hierarchy and add edges from each parent to each child. Allows us to remove redundant or overly specific categorization while preserving the overall structure.

- *Insert node above*: Add a dummy parent to push a node further down the hierarchy. Allows us to ensure that nodes of similar granularity are at the same level.

- *Delete node*: Remove a node and all of its edges. Used to remove abstract nodes that do not reveal visual characteristics.

- *Add edge*: Connect a node to a parent. Used to reassign the children of nodes deleted by the operation above.

We manually examined the hierarchy and implemented these actions in order to produce superclasses that are calibrated for classification. The resulting hierarchy contains nodes of comparable granularity at the same level. Moreover, as a result of this process, each node ends up having a single parent and thus the resulting hierarchy is a tree. The full hierarchy can be explored using the notebooks provided with the hierarchy in the Supplementary Material.

## A.5 RESULTING HIERARCHY

The parameters for constructing the BREEDS benchmarks (hierarchy level, number of subclasses, and tree root) are given in Table 2. The resulting tasks—obtained by sampling disjoint ImageNet classes (i.e., subpopulations) for the source and target domain—are shown in Tables 8, 9, 10, and 11. Recall that for each superclass we randomly sample a fixed number of subclasses per superclass to ensure that the dataset is approximately balanced.

| Superclass | Source | Target |
|---|---|---|
| **garment** | trench coat, abaya, gown, poncho, military uniform, jersey, cloak, bikini, miniskirt, swimming trunks | lab coat, brassiere, hoopskirt, cardigan, pajama, academic gown, apron, diaper, sweatshirt, sarong |
| **bird** | African grey, bee eater, coucal, American coot, indigo bunting, king penguin, spoonbill, limpkin, quail, kite | prairie chicken, red-breasted merganser, albatross, water ouzel, goose, oystercatcher, American egret, hen, lorikeet, ruffed grouse |
| **reptile** | Gila monster, agama, triceratops, African chameleon, thunder snake, Indian cobra, green snake, mud turtle, water snake, loggerhead | sidewinder, leatherback turtle, boa constrictor, garter snake, terrapin, box turtle, ringneck snake, rock python, American chameleon, green lizard |
| **arthropod** | rock crab, black and gold garden spider, tiger beetle, black widow, barn spider, leafhopper, ground beetle, fiddler crab, bee, walking stick | cabbage butterfly, admiral, lacewing, trilobite, sulphur butterfly, cicada, garden spider, leaf beetle, long-horned beetle, fly |
| **mammal** | Siamese cat, ibex, tiger, hippopotamus, Norwegian elkhound, dugong, colobus, Samoyed, Persian cat, Irish wolfhound | English setter, llama, lesser panda, armadillo, indri, giant schnauzer, pug, Doberman, American Staffordshire terrier, beagle |
| **accessory** | bib, feather boa, stole, plastic bag, bathing cap, cowboy boot, necklace, crash helmet, gasmask, maillot | hair slide, umbrella, pickelhaube, mitten, sombrero, shower cap, sock, running shoe, mortarboard, handkerchief |
| **craft** | catamaran, speedboat, fireboat, yawl, airliner, container ship, liner, trimaran, space shuttle, aircraft carrier | schooner, gondola, canoe, wreck, warplane, balloon, submarine, pirate, lifeboat, airship |
| **equipment** | volleyball, notebook, basketball, handheld computer, tripod, projector, barbell, monitor, croquet ball, balance beam | cassette player, snorkel, horizontal bar, soccer ball, racket, baseball, joystick, microphone, tape player, reflex camera |
| **furniture** | wardrobe, toilet seat, file, mosquito net, four-poster, bassinet, chiffonier, folding chair, fire screen, shoji | studio couch, throne, crib, rocking chair, dining table, park bench, chest, window screen, medicine chest, barber chair |
| **instrument** | upright, padlock, lighter, steel drum, parking meter, cleaver, syringe, abacus, scale, corkscrew | maraca, saltshaker, magnetic compass, accordion, digital clock, screw, can opener, odometer, organ, screwdriver |
| **man-made structure** | castle, bell cote, fountain, planetarium, traffic light, breakwater, cliff dwelling, monastery, prison, water tower | suspension bridge, worm fence, turnstile, tile roof, beacon, street sign, maze, chainlink fence, bakery, drilling platform |
| **wheeled vehicle** | snowplow, trailer truck, racer, shopping cart, unicycle, motor scooter, passenger car, minibus, jeep, recreational vehicle | jinrikisha, golfcart, tow truck, ambulance, bullet train, fire engine, horse cart, streetcar, tank, Model T |
| **produce** | broccoli, corn, orange, cucumber, spaghetti squash, butternut squash, acorn squash, cauliflower, bell pepper, fig | pomegranate, mushroom, strawberry, lemon, head cabbage, Granny Smith, hip, ear, banana, artichoke |

Table 8: Superclasses used for the ENTITY-13 task, along with the corresponding subpopulations that comprise the source and target domains.

| Superclass | Source | Target |
|---|---|---|
| **serpentes** | green mamba, king snake, garter snake, thunder snake | boa constrictor, green snake, ringneck snake, rock python |
| **passerine** | goldfinch, brambling, water ouzel, chickadee | magpie, house finch, indigo bunting, bulbul |
| **saurian** | alligator lizard, Gila monster, American chameleon, green lizard | Komodo dragon, African chameleon, agama, banded gecko |
| **arachnid** | harvestman, barn spider, scorpion, black widow | wolf spider, black and gold garden spider, tick, tarantula |
| **aquatic bird** | albatross, red-backed sandpiper, crane, white stork | goose, dowitcher, limpkin, drake |
| **crustacean** | crayfish, spiny lobster, hermit crab, Dungeness crab | king crab, rock crab, American lobster, fiddler crab |
| **carnivore** | Italian greyhound, black-footed ferret, Bedlington terrier, basenji | flat-coated retriever, otterhound, Shih-Tzu, Boston bull |
| **insect** | lacewing, fly, grasshopper, sulphur butterfly | long-horned beetle, leafhopper, dung beetle, admiral |
| **ungulate** | llama, gazelle, zebra, ox | hog, hippopotamus, hartebeest, warthog |
| **primate** | baboon, howler monkey, Madagascar cat, chimpanzee | siamang, indri, capuchin, patas |
| **bony fish** | coho, tench, lionfish, rock beauty | sturgeon, puffer, eel, gar |
| **barrier** | breakwater, picket fence, turnstile, bannister | chainlink fence, stone wall, dam, worm fence |
| **building** | bookshop, castle, mosque, butcher shop | grocery store, toyshop, palace, beacon |
| **electronic equipment** | printer, pay-phone, microphone, computer keyboard | modem, cassette player, monitor, dial telephone |
| **footwear** | clog, Loafer, maillot, running shoe | sandal, knee pad, cowboy boot, Christmas stocking |
| **garment** | academic gown, apron, miniskirt, fur coat | jean, vestment, sarong, swimming trunks |
| **headdress** | pickelhaube, hair slide, shower cap, bonnet | bathing cap, cowboy hat, bearskin, crash helmet |
| **home appliance** | washer, microwave, Crock Pot, vacuum | toaster, espresso maker, space heater, dishwasher |
| **kitchen utensil** | measuring cup, cleaver, coffeepot, spatula | frying pan, cocktail shaker, tray, caldron |
| **measuring instrument** | digital watch, analog clock, parking meter, magnetic compass | barometer, wall clock, hourglass, digital clock |
| **motor vehicle** | limousine, school bus, moped, convertible | trailer truck, beach wagon, police van, garbage truck |
| **musical instrument** | French horn, maraca, grand piano, upright | acoustic guitar, organ, electric guitar, violin |
| **neckwear** | feather boa, neck brace, bib, Windsor tie | necklace, stole, bow tie, bolo tie |

| | | |
|---|---|---|
| **sports equipment** | ski, dumbbell, croquet ball, racket | rugby ball, balance beam, horizontal bar, tennis ball |
| **tableware** | mixing bowl, water jug, beer glass, water bottle | goblet, wine bottle, coffee mug, plate |
| **tool** | quill, combination lock, padlock, screw | fountain pen, screwdriver, shovel, torch |
| **vessel** | container ship, lifeboat, aircraft carrier, trimaran | liner, wreck, catamaran, yawl |
| **dish** | potpie, mashed potato, pizza, cheeseburger | burrito, hot pot, meat loaf, hotdog |
| **vegetable** | zucchini, cucumber, butternut squash, artichoke | cauliflower, spaghetti squash, acorn squash, cardoon |
| **fruit** | strawberry, pineapple, jackfruit, Granny Smith | buckeye, corn, ear, acorn |

Table 9: Superclasses used for the ENTITY-30 task, along with the corresponding subpopulations that comprise the source and target domains.

| Superclass | Source | Target |
|---|---|---|
| **salamander** | eft, axolotl | common newt, spotted salamander |
| **turtle** | box turtle, leatherback turtle | loggerhead, mud turtle |
| **lizard** | whiptail, alligator lizard | African chameleon, banded gecko |
| **snake** | night snake, garter snake | sea snake, boa constrictor |
| **spider** | tarantula, black and gold garden spider | garden spider, wolf spider |
| **grouse** | ptarmigan, prairie chicken | ruffed grouse, black grouse |
| **parrot** | macaw, lorikeet | African grey, sulphur-crested cockatoo |
| **crab** | Dungeness crab, fiddler crab | rock crab, king crab |
| **dog** | bloodhound, Pekinese | Great Pyrenees, papillon |
| **wolf** | coyote, red wolf | white wolf, timber wolf |
| **fox** | grey fox, Arctic fox | red fox, kit fox |
| **domestic cat** | tiger cat, Egyptian cat | Persian cat, Siamese cat |
| **bear** | sloth bear, American black bear | ice bear, brown bear |
| **beetle** | dung beetle, rhinoceros beetle | ground beetle, long-horned beetle |
| **butterfly** | sulphur butterfly, admiral | cabbage butterfly, ringlet |
| **ape** | gibbon, orangutan | gorilla, chimpanzee |
| **monkey** | marmoset, titi | spider monkey, howler monkey |

Table 10: Superclasses used for the LIVING-17 task, along with the corresponding subpopulations that comprise the source and target domains.

| Superclass | Source | Target |
|---|---|---|
| **bag** | plastic bag, purse | mailbag, backpack |
| **ball** | volleyball, punching bag | ping-pong ball, soccer ball |
| **boat** | gondola, trimaran | catamaran, canoe |
| **body armor** | bulletproof vest, breastplate | chain mail, cuirass |
| **bottle** | pop bottle, beer bottle | wine bottle, water bottle |
| **bus** | trolleybus, minibus | school bus, recreational vehicle |
| **car** | racer, Model T | police van, ambulance |
| **chair** | folding chair, throne | rocking chair, barber chair |
| **coat** | lab coat, fur coat | kimono, vestment |
| **digital computer** | laptop, desktop computer | notebook, hand-held computer |
| **dwelling** | palace, monastery | mobile home, yurt |
| **fence** | worm fence, chainlink fence | stone wall, picket fence |
| **hat** | bearskin, bonnet | sombrero, cowboy hat |
| **keyboard instrument** | grand piano, organ | upright, accordion |
| **mercantile establishment** | butcher shop, barbershop | shoe shop, grocery store |
| **outbuilding** | greenhouse, apiary | barn, boathouse |
| **percussion instrument** | steel drum, marimba | drum, gong |
| **pot** | teapot, Dutch oven | coffeepot, caldron |
| **roof** | dome, vault | thatch, tile roof |
| **ship** | schooner, pirate | aircraft carrier, liner |
| **skirt** | hoopskirt, miniskirt | overskirt, sarong |
| **stringed instrument** | electric guitar, banjo | violin, acoustic guitar |
| **timepiece** | digital watch, stopwatch | parking meter, digital clock |
| **truck** | fire engine, pickup | tractor, forklift |
| **wind instrument** | oboe, sax | flute, bassoon |
| **squash** | spaghetti squash, acorn squash | zucchini, butternut squash |

Table 11: Superclasses used for the NON-LIVING-26 task, along with the corresponding subpopulations that comprise the source and target domains.

## A.6    ANNOTATOR TASK

As described in Section 4.3, the goal of our human studies is to understand whether humans can classify images into superclasses even without knowing the semantic grouping. Thus, the task involved showing annotators two groups of images, each sampled from the source domain of a random superclass. Then, annotators were shown a new set of images from the target domain (or the source domain in the case of control) and were asked to assign each of them into one of the two groups. A screenshot of an (random) instance of our annotator task is shown in Figure 12.

Each task contained 20 images from the source domain of each superclass and 12 images for annotators to classify (the images where rescaled and center-cropped to size $224 \times 224$ to match the input size use for model predictions). The two superclasses were randomly permuted at load time. To ensure good concentration of our accuracy estimates, for every superclass, we performed binary classification tasks w.r.t. 3 other (randomly chosen) superclasses. Further, we used 3 annotators per task and annotators were compensated $0.15 per task.

**Comparing with the original hierarchy.**    In order to compare our superclasses with those obtained by Huh et al. (2016) via WordNet clustering,[4] we need to define a correspondence between them. To do so, for each of our tasks, we selected the clustering (either top-down or bottom-up) that had the closest number of superclasses. Following the terminology from that work, this mapping is: ENTITY-13 → DOWNUP-36, ENTITY-30 → UPDOWN-127, LIVING-17 → DOWNUP-753 (restricted to "living" nodes), and NON-LIVING-26 → DOWNUP-345 (restricted to "non-living" nodes).

---

[4]https://github.com/minyoungg/wmigftl/tree/master/label_sets/hierarchy

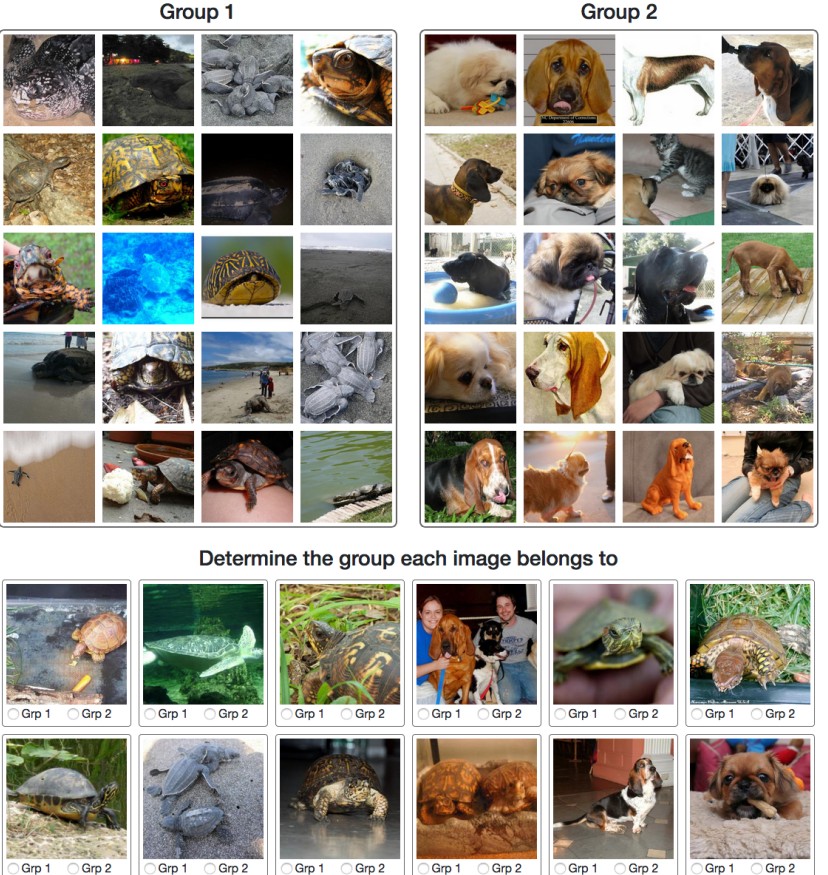

Figure 12: Sample MTurk annotation task to obtain human baselines for BREEDS benchmarks.

### A.7 EVALUATING MODEL PERFORMANCE

#### A.7.1 MODEL ARCHITECTURES AND TRAINING

The model architectures used in our analysis are in Table 13 for which we used standard implementations from the PyTorch library (`https://pytorch.org/docs/stable/torchvision/models.html`). For training, we use a batch size of 128, weight decay of $10^{-4}$, and learning rates listed in Table 13. Models were trained until convergence. On ENTITY-13 and ENTITY-30, this required a total of 300 epochs, with 10-fold drops in learning rate every 100 epochs, while on LIVING-17and NON-LIVING-26, models a total of 450 epochs, with 10-fold learning rate drops every 150 epochs. For adapting models, we retrained the last (fully-connected) layer on the train split of the target domain, starting from the parameters of the source-trained model. We trained that layer using SGD with a batch size of 128 for 40,000 steps and chose the best learning rate out of $[0.01, 0.1, 0.25, 0.5, 1.0, 2.0, 3.0, 5.0, 7.0, 8.0, 10.0, 11.0, 12.0]$, based on test accuracy.

| Model | Learning Rate |
|---|---|
| alexnet | 0.01 |
| vgg11 | 0.01 |
| resnet18 | 0.1 |
| resnet34 | 0.1 |
| resnet50 | 0.1 |
| densenet121 | 0.1 |

Table 13: Models used in our analysis.

#### A.7.2 MODEL PAIRWISE ACCURACY

In order to make a fair comparison between the performance of models and human annotators on the BREEDS tasks, we evaluate model accuracy on pairs of superclasses. On images from that pair, we determine the model prediction to be the superclass for which the model's predicted probability is higher. A prediction is deemed correct if it matches the superclass label for the image. Repeating this process over random pairs of superclasses allows us to estimate model accuracy on the average-case binary classification task.

#### A.7.3 ROBUSTNESS INTERVENTIONS

For model training, we use the hyperparameters provided in Appendix A.7.1. Additional intervention-specific hyperparameters are listed in Appendix Table 14. Due to computational constraints, we trained a restricted set of model architectures with robustness interventions—ResNet-18 and ResNet-50 for adversarial training, and ResNet-18 and ResNet-34 for all others. Adversarial training was implemented using the `robustness` library,[5] while random erasing using the PyTorch `transforms`.[6]

| Eps | Step size | #Steps | | Mean | StdDev | | Probability | Scale | Ratio |
|---|---|---|---|---|---|---|---|---|---|
| 0.5 | 0.4 | 3 | | | | | | | |
| 1 | 0.8 | 3 | | 0 | 0.2 | | 0.5 | 0.02 - 0.33 | 0.3 - 3.3 |

(a) PGD-training (Madry et al., 2018)    (b) Gaussian noise    (c) Random erasing

Table 14: Additional hyperparameters for robustness interventions.

---

[5] `https://github.com/MadryLab/robustness`
[6] `https://pytorch.org/docs/stable/torchvision/transforms.html`

# B    ADDITIONAL RELATED WORK

In Section 2, we provide an overview of prior work that is focused on evaluating model robustness to distribution shift. In Section 6, we discuss existing benchmarks that are most similar to our work. Here, we discuss other research direction related to model robustness and generalization.

**Distributional robustness.**    Distribution shifts that are small with respect to some $f$-divergence have been studied in prior theoretical work (Ben-Tal et al., 2013; Duchi et al., 2016; Esfahani & Kuhn, 2018; Namkoong & Duchi, 2016). However, this notion of robustness is typically too pessimistic to capture realistic data variations (Hu et al., 2018). Distributional robustness has also been connected to causality (Meinshausen, 2018): here, the typical approach is to inject spurious correlations into the dataset, and assess to what extent models rely on them for their predictions (Heinze-Deml & Meinshausen, 2017; Arjovsky et al., 2019; Sagawa et al., 2020).

**Domain adaptation and transfer learning.**    The goal here is to adapt models to the target domain with relatively few samples from it (Ben-David et al., 2007; Saenko et al., 2010; Ganin & Lempitsky, 2015; Courty et al., 2016; Gong et al., 2016; Donahue et al., 2014; Sharif Razavian et al., 2014). In domain adaptation, the task is the same in both domains, while in transfer learning, the task itself could vary. In a similar vein, the field of *domain generalization* aims to generalize to samples from a different domain (e.g., from ClipArt to photos) by training on a number of explicitly annotated domains (Muandet et al., 2013; Li et al., 2017; Peng et al., 2019).

**Zero-shot learning.**    Work in this domain focuses on learning to recognize previously unseen classes (Lampert et al., 2009; Xian et al., 2017), typically described via a semantic embedding (Lampert et al., 2009; Mikolov et al., 2013; Socher et al., 2013; Frome et al., 2013; Romera-Paredes & Torr, 2015). This differs from our setup, where the focus is on generalization to unseen subpopulations for the *same* set of classes.

# C  ADDITIONAL EXPERIMENTAL RESULTS

## C.1  HUMAN BASELINES FOR BREEDS TASKS

In Section 4.3, we evaluate human performance on binary versions of our BREEDS tasks. Appendix Figures 15a and 15b show the distribution of annotator accuracy over different pairs of superclasses for test data sampled from the source and target domains respectively.

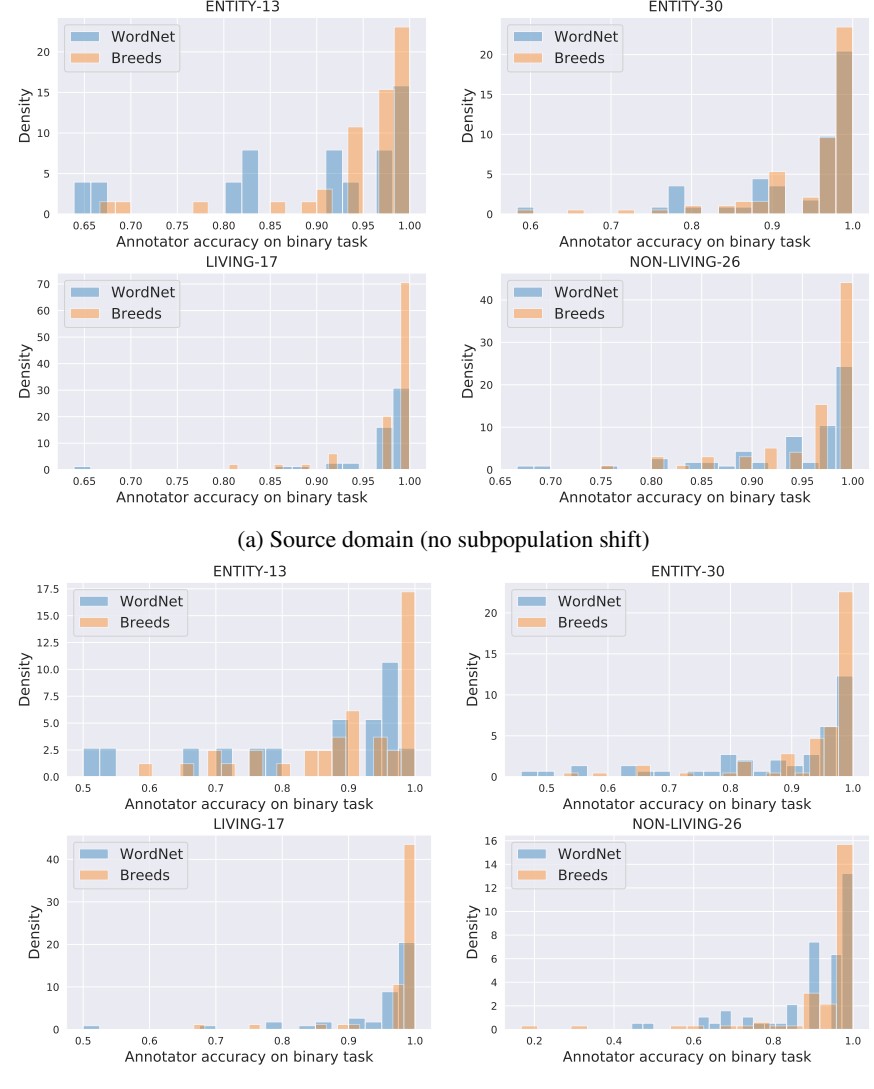

Figure 15: Distribution of annotator accuracy over pairwise superclass classification tasks. We observe that human annotators consistently perform better on tasks constructed using our modified ImageNet class hierarchy (i.e., BREEDS) as opposed to those obtained directly from WordNet.

## C.2  Model Evaluation

In Figures 16- 18, we visualize model performance over BREEDS superclasses for different model architectures. We observe in general that models perform fairly uniformly over classes when the test data is drawn from the source domain. This indicates that the tasks are well-calibrated—the various superclasses are of comparable difficulty. At the same time, we see that model robustness to subpopulation shift, i.e., drop in accuracy on the target domain, varies widely over superclasses. This could be either due to some superclasses being broader by construction or due to models being more sensitive to subpopulation shift for some classes.

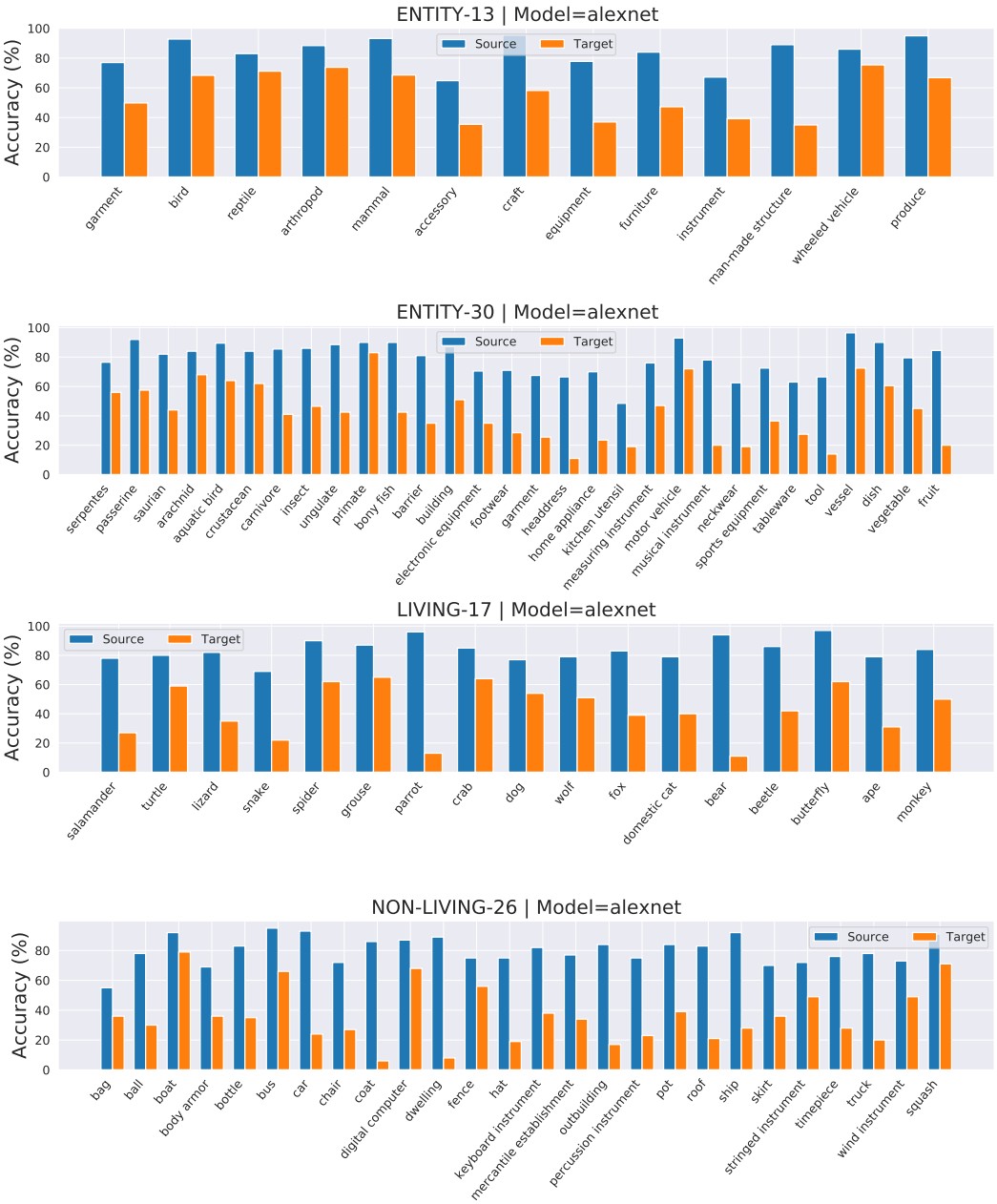

Figure 16: Per-class source and target accuracies for AlexNet on BREEDS tasks.

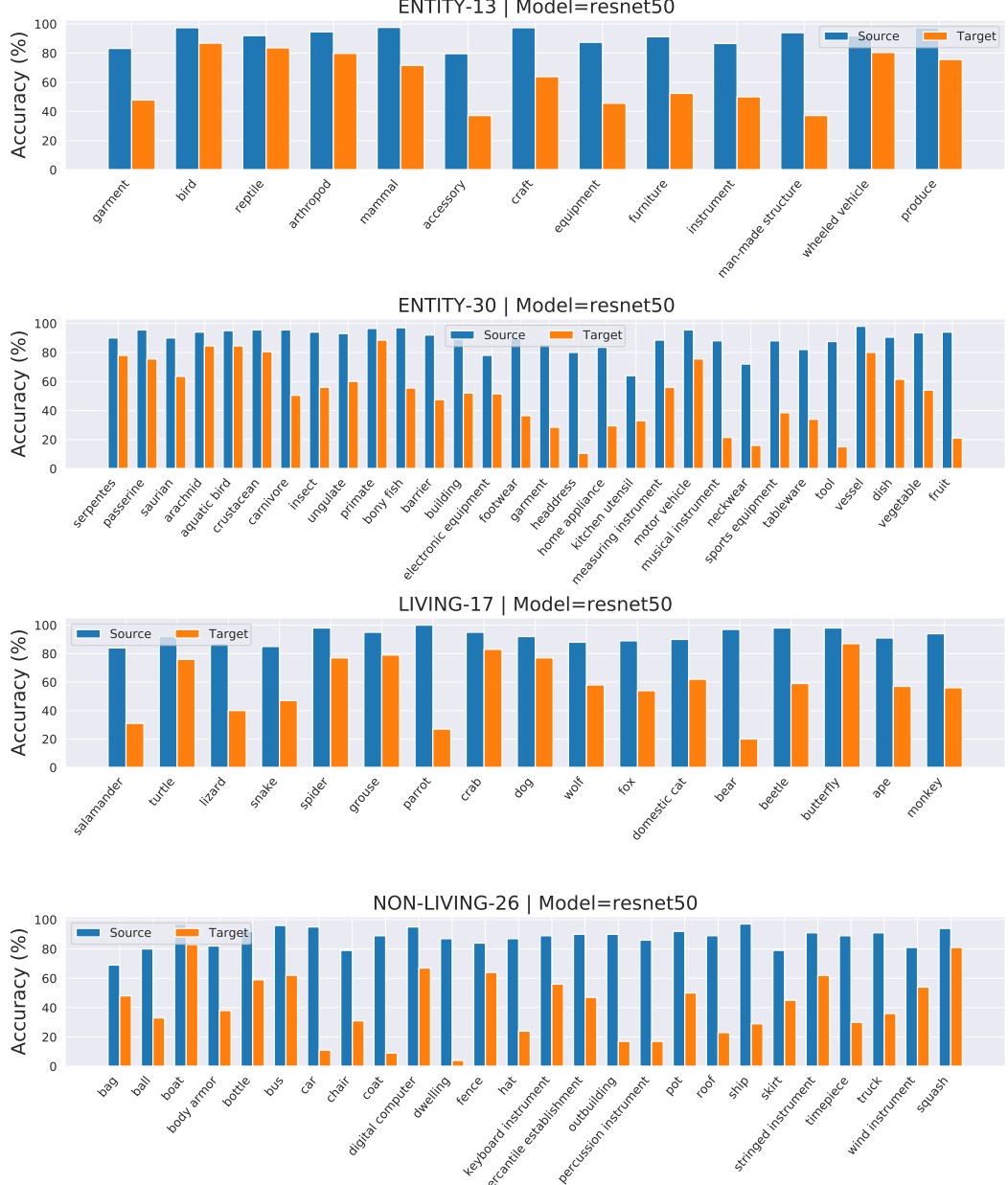

Figure 17: Per-class source and target accuracies for ResNet-50 on BREEDS tasks.

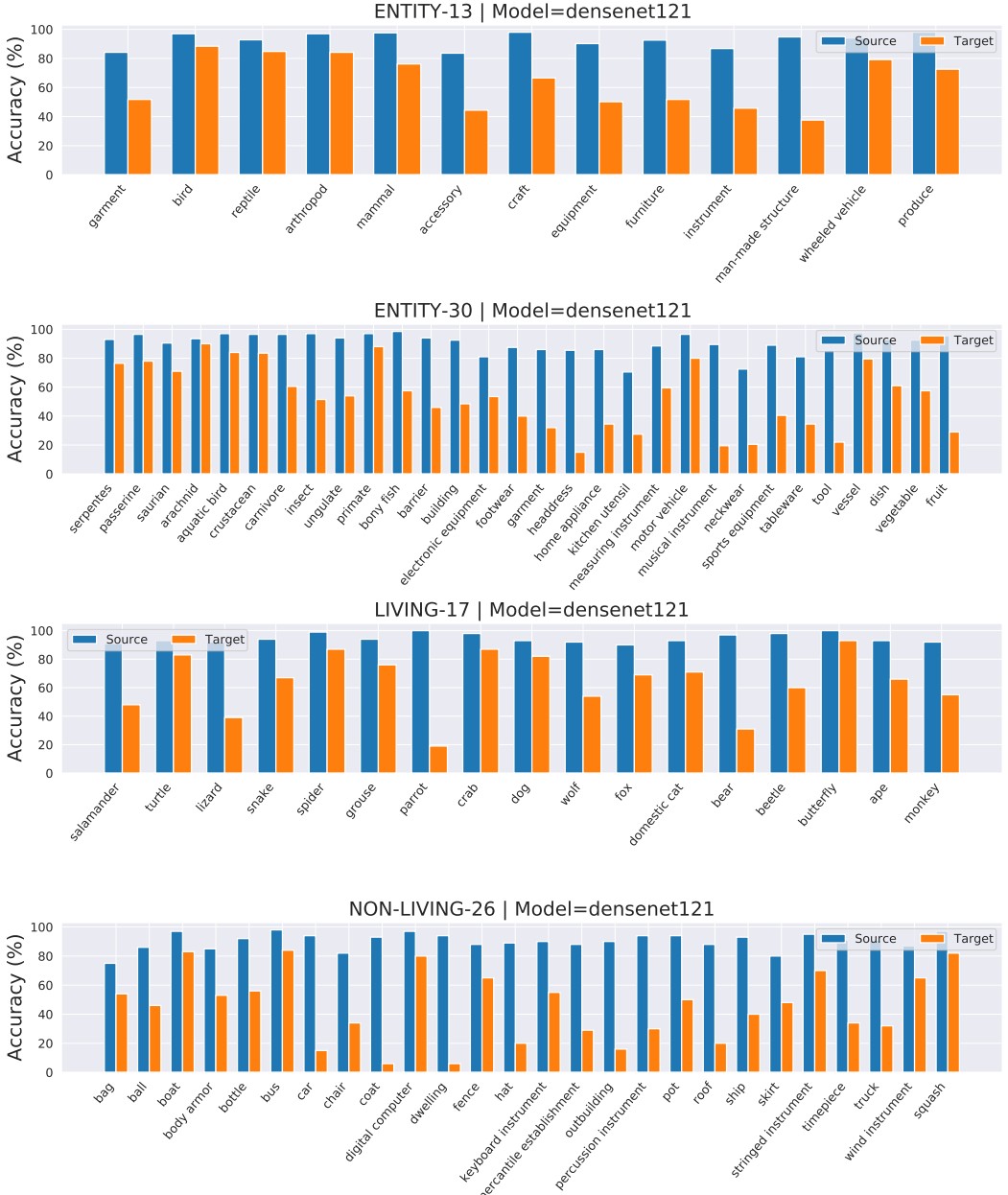

Figure 18: Per-class source and target accuracies for DenseNet-121 on BREEDS tasks.

### C.2.1 EFFECT OF DIFFERENT SPLITS

As described in Section 3, to create BREEDS tasks, we first identify a set of relevant superclasses (at the chosen depth in the hierarchy), and then partition their subpopulations between the source and target domains. For all the tasks listed in Table 2, the superclasses are balanced—each of them comprise the same number of subpopulations. To ensure this is the case, the desired number of subpopulations is chosen among all superclass subpopulations at random. These subpopulations are then randomly split between the source and target domains.

Instead of randomly partitioning subpopultions (of a given superclass) between the two domains, we could instead craft partitions to be more/less adversarial as illustrated in Figure 19. Specifically, we could control how similar the subpopulations in the target domain are to those in the source domain. For instance, a split would be less adversarial (*good*) if subpopulations in the source and target domain share a common parent. On the other hand, we could make a split more adversarial (*bad*) by ensuring a greater degree of separation (in terms of distance in the hierarchy) between the source and target domain subpopulations.

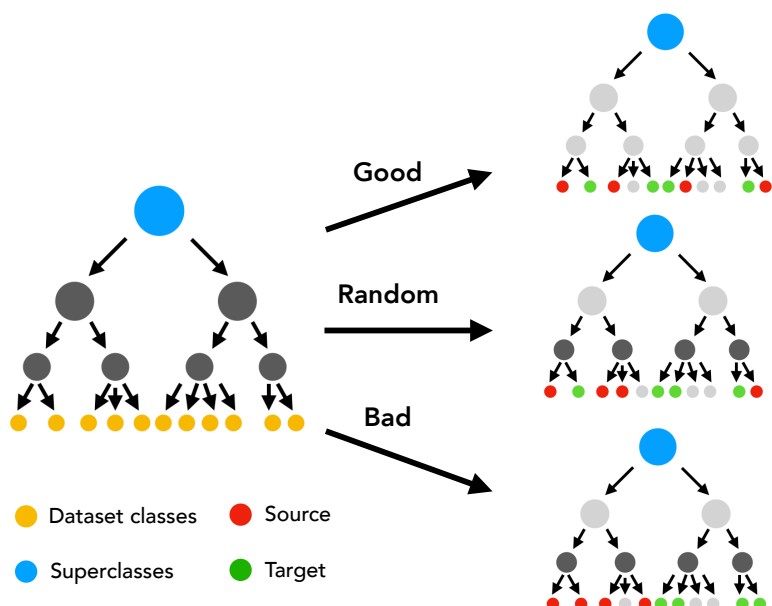

Figure 19: Different ways to partition the subpopulations of a given superclass into the source and target domains. Depending on how closely related the subpopulations in the two domain are, we can construct splits that are more/less adversarial.

We now evaluate model performance under such variations in the nature of the splits themselves—see Figure 20. As expected, models perform comparably well on test data from the source domain, independent of the how the subpopulations are partitioned into the two domains. However, model robustness to subpopulation shift varies considerably based on the nature of the split—it is lowest for the most adversarially chosen split. Finally, we observe that retraining the linear layer on data from the target domain recovers a considerable fraction of the accuracy drop in all cases—indicating that even for the more adversarial splits, models do learn features that transfer well to unknown subpopulations.

### C.2.2 ROBUSTNESS INTERVENTIONS

In Tables 22 and 23, we present the raw accuracies of models trained using various train-time robustness interventions.

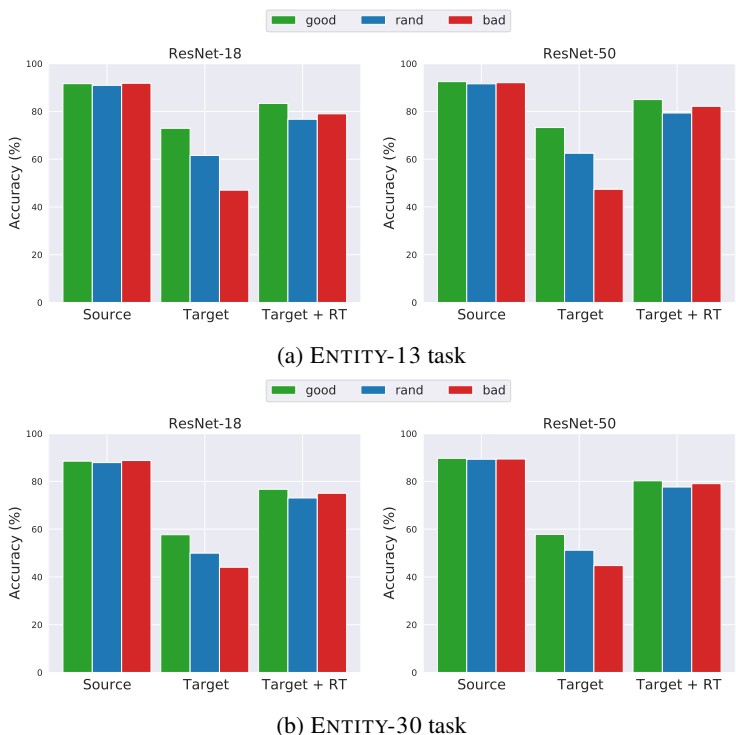

Figure 20: Model robustness as a function of the nature of subpopulation shift within specific BREEDS tasks. We vary how the underlying subpopulations of each superclass are split between the source and target domain—we compare random splits (used in the majority of our analysis), to ones that are more (*bad*) or less adversarial (*good*). When models are tested on samples from the source domain, they perform equally well across different splits, as one might expect. However, under subpopulation shift (i.e., on samples from the target domain), model robustness varies drastically, and is considerably worse when the split is more adversarial. Yet, for all the splits, models have comparable target accuracy after retraining their final layer.

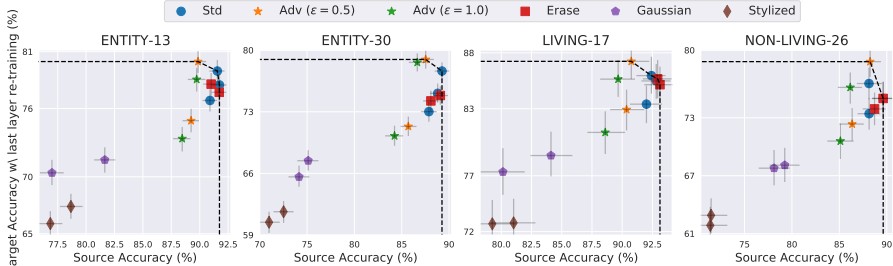

Figure 21: Target accuracy of models after they have been retrained (only the final linear layer) on data from the target domain (with 95% bootstrap confidence intervals). Models trained with robustness interventions often have higher target accuracy than standard models post retraining.

| ResNet-18 | | | | |
|---|---|---|---|---|
| Task | $\varepsilon$ | Source | Accuracy (%) Target | Target-RT |
| ENTITY-13 | 0 | **90.91 ± 0.73** | **61.52 ± 1.23** | **76.71 ± 1.09** |
| | 0.5 | 89.23 ± 0.80 | **61.10 ± 1.23** | 74.92 ± 1.04 |
| | 1.0 | 88.45 ± 0.81 | 58.53 ± 1.26 | 73.35 ± 1.11 |
| ENTITY-30 | 0 | **87.88 ± 0.89** | **49.96 ± 1.31** | **73.05 ± 1.17** |
| | 0.5 | 85.68 ± 0.91 | **48.93 ± 1.34** | 71.34 ± 1.14 |
| | 1.0 | 84.23 ± 0.91 | 47.66 ± 1.23 | 70.27 ± 1.17 |
| LIVING-17 | 0 | **92.01 ± 1.30** | **58.21 ± 2.32** | **83.38 ± 1.79** |
| | 0.5 | 90.35 ± 1.35 | 55.79 ± 2.44 | **83.00 ± 1.89** |
| | 1.0 | 88.56 ± 1.50 | 53.89 ± 2.36 | 80.90 ± 1.92 |
| NON-LIVING-26 | 0 | **88.09 ± 1.28** | **41.87 ± 2.01** | **73.52 ± 1.71** |
| | 0.5 | 86.28 ± 1.32 | **41.02 ± 1.91** | **72.41 ± 1.71** |
| | 1.0 | 85.19 ± 1.38 | **40.23 ± 1.92** | 70.61 ± 1.73 |

| ResNet-50 | | | | |
|---|---|---|---|---|
| Task | $\varepsilon$ | Source | Accuracy (%) Target | Target-RT |
| ENTITY-13 | 0 | **91.54 ± 0.64** | **62.48 ± 1.16** | **79.32 ± 1.01** |
| | 0.5 | 89.87 ± 0.80 | **63.01 ± 1.15** | **80.14 ± 1.00** |
| | 1.0 | 89.71 ± 0.74 | 61.21 ± 1.22 | 78.58 ± 0.98 |
| ENTITY-30 | 0 | **89.26 ± 0.78** | **51.18 ± 1.24** | 77.60 ± 1.17 |
| | 0.5 | 87.51 ± 0.88 | **50.72 ± 1.28** | **78.92 ± 1.06** |
| | 1.0 | 86.63 ± 0.88 | **50.99 ± 1.27** | **78.63 ± 1.03** |
| LIVING-17 | 0 | **92.40 ± 1.28** | **58.22 ± 2.42** | **85.96 ± 1.72** |
| | 0.5 | 90.79 ± 1.55 | 55.97 ± 2.38 | **87.22 ± 1.66** |
| | 1.0 | 89.64 ± 1.47 | 54.64 ± 2.48 | **85.63 ± 1.73** |
| NON-LIVING-26 | 0 | **88.13 ± 1.30** | **41.82 ± 1.86** | 76.58 ± 1.69 |
| | 0.5 | **88.20 ± 1.20** | **42.57 ± 2.03** | **78.84 ± 1.62** |
| | 1.0 | 86.17 ± 1.36 | **41.69 ± 1.96** | 76.16 ± 1.61 |

Table 22: Effect of adversarial training on model robustness to subpopulation shift. All models are trained on samples from the source domain—either using standard training ($\varepsilon = 0.0$) or using adversarial training. Models are then evaluated in terms of: (a) source accuracy, (b) target accuracy and (c) target accuracy after retraining the linear layer of the model with data from the target domain. Confidence intervals (95%) obtained via bootstrapping. Maximum task accuracy over $\varepsilon$ (taking into account confidence interval) shown in bold.

| | | Accuracy (%) | | |
|---|---|---|---|---|
| | | ResNet-18 | | |
| Task | Intervention | Source | Target | Target-RT |
| ENTITY-13 | Standard | **90.91 ± 0.73** | **61.52 ± 1.23** | 76.71 ± 1.09 |
| | Erase Noise | **91.01 ± 0.68** | **62.79 ± 1.27** | **78.10 ± 1.09** |
| | Gaussian Noise | 77.00 ± 1.04 | 47.90 ± 1.21 | 70.37 ± 1.17 |
| | Stylized ImageNet | 76.85 ± 1.00 | 50.18 ± 1.21 | 65.91 ± 1.17 |
| ENTITY-30 | Standard | **87.88 ± 0.89** | **49.96 ± 1.31** | 73.05 ± 1.17 |
| | Erase Noise | **88.09 ± 0.80** | **49.98 ± 1.31** | **74.27 ± 1.15** |
| | Gaussian Noise | 74.12 ± 1.16 | 35.79 ± 1.21 | 65.62 ± 1.28 |
| | Stylized ImageNet | 70.96 ± 1.16 | 37.67 ± 1.21 | 60.45 ± 1.22 |
| LIVING-17 | Standard | **92.01 ± 1.30** | **58.21 ± 2.32** | **83.38 ± 1.79** |
| | Erase Noise | **93.09 ± 1.27** | **59.60 ± 2.40** | **85.12 ± 1.71** |
| | Gaussian Noise | 80.13 ± 1.99 | 46.16 ± 2.57 | 77.31 ± 2.08 |
| | Stylized ImageNet | 79.21 ± 1.85 | 43.96 ± 2.38 | 72.74 ± 2.09 |
| NON-LIVING-26 | Standard | **88.09 ± 1.28** | **41.87 ± 2.01** | **73.52 ± 1.71** |
| | Erase Noise | **88.68 ± 1.18** | **43.17 ± 2.10** | **73.91 ± 1.78** |
| | Gaussian Noise | 78.14 ± 1.60 | 35.13 ± 1.94 | 67.79 ± 1.79 |
| | Stylized ImageNet | 71.43 ± 1.73 | 30.56 ± 1.75 | 61.83 ± 1.98 |

| | | Accuracy (%) | | |
|---|---|---|---|---|
| | | ResNet-34 | | |
| Task | Intervention | Source | Target | Target-RT |
| ENTITY-13 | Standard | **91.75 ± 0.70** | **63.45 ± 1.13** | **78.07 ± 1.02** |
| | Erase Noise | **91.76 ± 0.70** | **62.71 ± 1.25** | **77.43 ± 1.06** |
| | Gaussian Noise | 81.60 ± 0.97 | 50.69 ± 1.28 | 71.50 ± 1.13 |
| | Stylized ImageNet | 78.66 ± 0.94 | 51.05 ± 1.30 | 67.38 ± 1.16 |
| ENTITY-30 | Standard | **88.81 ± 0.81** | **51.68 ± 1.28** | **75.12 ± 1.11** |
| | Erase Noise | **89.07 ± 0.82** | **51.04 ± 1.27** | **74.88 ± 1.08** |
| | Gaussian Noise | 75.05 ± 1.11 | 38.31 ± 1.26 | 67.47 ± 1.22 |
| | Stylized ImageNet | 72.51 ± 1.10 | 38.98 ± 1.22 | 61.65 ± 1.25 |
| LIVING-17 | Standard | **92.83 ± 1.19** | 59.74 ± 2.27 | **85.46 ± 1.83** |
| | Erase Noise | **92.96 ± 1.32** | **61.13 ± 2.30** | **85.66 ± 1.78** |
| | Gaussian Noise | 84.06 ± 1.71 | 48.38 ± 2.44 | 78.79 ± 1.91 |
| | Stylized ImageNet | 80.94 ± 2.00 | 44.16 ± 2.43 | 72.77 ± 2.18 |
| NON-LIVING-26 | Standard | **89.64 ± 1.17** | **43.03 ± 1.99** | **74.99 ± 1.66** |
| | Erase Noise | **89.62 ± 1.31** | **43.53 ± 1.89** | **75.04 ± 1.70** |
| | Gaussian Noise | 79.26 ± 1.61 | 34.89 ± 1.91 | 68.07 ± 1.78 |
| | Stylized ImageNet | 71.49 ± 1.65 | 31.10 ± 1.80 | 62.94 ± 1.90 |

Table 23: Effect of various train-time interventions on model robustness to subpopulation shift. All models are trained on samples from the the source domain. Models are then evaluated in terms of: (a) source accuracy, (b) target accuracy and (c) target accuracy after retraining the linear layer of the model with data from the target domain. Confidence intervals (95%) obtained via bootstrapping. Maximum task accuracy over $\varepsilon$ (taking into account confidence interval) shown in bold.

