# OpenReview forum: "BREEDS: Benchmarks for Subpopulation Shift"
_ICLR.cc/2021/Conference — ICLR 2021 Poster_

### Official Review · AnonReviewer1 · 2020-10-14
**Straightforward approach for evaluating model robustness to subpopulation shift**

**Rating:** 6
**Confidence:** 3

**Review:**

This paper addresses the problem of model robustness to subpopulation shift. Authors propose building large-scale subpopulation shift benchmarks wherein the data subpopulations present during model training and evaluation differ. In this regard, their approach is based on leveraging existing dataset labels and use them to identify superclasses to construct classification tasks over such superclasses and repurpose the original dataset classes to be the subpopulations of interest. They train some learning models over the generated benchmarks to evaluate model robustness to subpopulation shift and, finally, they try various learning interventions (from the literature) to decrease model sensitivity to this sort of data perturbations.


Strengths:

- Paper is very well-written.
- The problem addressed is very important (learning model generalisation to data shift) and of interest for the majority of ML/AI research community.
- The methodology followed is well defined and correct.
- The authors have performed an excellent work with the comprehensive experimental setting proposed in the paper.

Weaknesses:

- It is difficult to characterize what new scientific understanding or knowledge was presented in this paper. The presented approach for identifying superclasses and subpopulations of interests is somehow straightforward. Also, I doubt that manual procedure for hierarchy creation / restructuring process is a trivial task (with relatively little effort) for most benchmarks with no structured organisation. Results in sections 4.3 and 5.1 appear entirely unsurprising (though admittedly, this could be hindsight bias). Results in 5.2 appear to present more interesting insights (e.g.  little effect of train-time interventions on model robustness to subpopulation shift). However, one is left wondering whether this insight generalizes beyond the specifics of this experiments/dataset or whether this will create an isolated ImageNet sub-community for addressing image-classification robustness tasks.


Comments after author rebuttal:

Looking at the author's comments (as well as the other reviewer's feedback), I think that the authors have made a good job with the responses and I'm now more convinced about the usefulness of this work. I'm increasing my original recommendation to 6 "Marginally above threshold".

---

> ### Author Response · Authors · 2020-11-21
> **Author response**
>
> We thank the reviewer for their kind comments about our work.
>
> **[Scientific understanding]** Although robustness to distribution shift is widely accepted as an important problem in the community, we still lack a coherent methodology (and the associated large-scale benchmarks) to actually assess model vulnerabilities and design interventions. This is particularly true in the context of subpopulation shifts, for which few (and rather limited) benchmarks exist. Even though the insight of leveraging class structure to create such benchmarks may seem natural in retrospect, we are not aware of any work that actually applies this methodology to study distribution shift. Moreover, we view the simplicity of our approach as a strength---it allows us to semi-automatically create a suite of benchmarks without any additional data collection or synthetic modeling that we believe will be useful for developing similar benchmarks in the future.
>
> **[Manually restructuring the hierarchy]** While we do agree that calibrating the class hierarchy is not trivial in terms of manual effort, it is significantly less tedious and expensive than collecting a new dataset with subpopulation annotations from scratch. In particular, our approach enables us to restrict our focus only to the dataset classes, instead of annotating individual data points (which are orders of magnitude more). It also allows us to repurpose existing widely-used datasets for which subpopulation annotations are absent without _any_ additional data collection. Finally, note that there exist approaches that leverage crowdsourcing (e.g., [this paper](https://las.inf.ethz.ch/files/sun15building.pdf)) or ML (e.g., [this paper](https://ai.stanford.edu/~rion/swn/)) to group semantically similar classes together, thus reducing the manual effort required.
>
> **[Our results]** We view the experiments in section 4.3 as a means to validate that our benchmarks do indeed capture something meaningful---i.e., realistic data variations which are fairly insignificant for humans but disproportionately hamper state-of-the-art models. While this might seem unsurprising in hindsight, we believe it is crucial for establishing the validity of our benchmarks. Our experiments in Section 5.1 further corroborate this and highlight that subpopulation robustness is a concrete axis along which standard models can be improved. Finally, we believe the finding that more accurate models are more robust to subpopulation shift is not obvious/expected---more accurate models could be more prone to overfitting to the distribution they are trained on and thus not generalize as well to a shifted distribution (e.g., see the literature on [adaptive overfitting](https://arxiv.org/abs/1411.2664)).
>
> **[Isolated sub-community]** We agree that ImageNet is a small part of ML research. At the same time, the benchmarks we present form a concrete and challenging goal for research on model robustness. Thus, we do not view our works as the single perfect robustness benchmark (such a benchmark is unlikely to exist), but as a first step towards benchmarks for developing subpopulation shift-robust models. In fact, we believe that defining such concrete benchmarks or testbeds for evaluation is the only way for making tangible progress on these questions. Finally, as we discussed in Section 4.2  and in the response to Reviewer 4, we believe that our methodology could be easily extended beyond ImageNet to other image classification tasks and even other domains.

---

### Official Review · AnonReviewer3 · 2020-10-29
**A new benchmark for evaluating model's generalization on subpopulation**

**Rating:** 7
**Confidence:** 4

**Review:**

##########################################################################

Summary:


The paper introduces a new benchmark, based on a subset of Image-Net, to evaluate how well a model generalizes to data subpopulations which are not observed during training. The key concept is to create breeds, which is done in a semi-automatic fashion -- it starts from a modified version of the WordNet, and then calibrated by human annotators.

##########################################################################

Reasons for score:


I tend to accept this work, because this benchmark is a good way to evaluate the model's generalization in terms of subpopulation. Though there were several datasets for domain generalization, I think the view of this benchmark is different and it can be a starting point for another direction.

##########################################################################

Pros:


1. The benchmark introduced in this paper is very interesting. Roughly speaking, it tries to investigate: if I train a model with the images of "British shorthair cat" only (plus some other classes' images of course), and now present it an image of "American curl", will the model at least tell me it is a "cat".

2. The effort on constructing this dataset and evaluating existing methods is significant.

##########################################################################

Cons:


I may disagree this benchmark can be viewed as an instance of domain generalization (DG). DG most commonly refers to evaluating a model trained with photo cat only can generalize to carton cat. This study still focus on one domain (one style of images). Therefore, the chosen interventions in Sec 5.2 may not be the best choices.

##########################################################################

---

> ### Author Response · Authors · 2020-11-21
> **Author response**
>
> We thank the reviewer for their encouraging words about our work.
>
> We do agree that our approach deviates from the standard DG setting, given that DG benchmarks typically focus on training models to generalize between image styles (e.g., from photos to drawings). We will revise our manuscript to clarify this.
>
> That being said, the interventions in Section 5.2 were explicitly chosen with our application (photo-to-photo generalization) in mind. Specifically, interventions such as noise augmentation, adversarial training, and data stylization are often used to make models robust to distribution shift present in natural images (real-world photos) such as common corruptions (e.g., fog and blur, see [this paper](https://arxiv.org/abs/1903.12261) and [this paper](https://arxiv.org/abs/1908.08016)) and temporal data shifts in videos (cf. [this paper](https://arxiv.org/abs/2007.00644)). From this perspective, the interventions we chose are, to the best of our knowledge, the canonical choices for generalizing between different distributions of photos.

---

### Official Review · AnonReviewer4 · 2020-11-01
**Important problem addressed, lack of formalization and distinction from the related work.**

**Rating:** 6
**Confidence:** 3

**Review:**

The authors develop a framework named BREEDS for studying population shift, putting it in their words, they address the problem of how well do models generalize to data subpopulations they have seen during training, in the specific domain of images, without altering the inputs or requiring new data. They propose to create large scale subpopulation shift benchmarks  to assess how models generalize beyond the diversity of the training examples.  The underlying idea is to identify superclasses from the dataset.
Breeds can be used as a tool to test and improve models and increase their generalization capabilities to distribution shift.
The problem is very important and the paper well written. However,  the applicability is limited to domains with class structure. The main application presented is images although the authors claim that the approach could generalize to other ares like nlp. Grouping similar classes may not be possible in many application and a discussion about it could have been helpful.
The lack of formalization of the framework makes the paper content hard to read and overall vague. However, with some effort, it is possible to get a clear idea of their contribution. Still a formalization would have been very desirable.
Figure 1 claims it illustrates the pipeline but it is not clear what the figure represents, there is no pipeline per se.

---

> ### Author Response · Authors · 2020-11-21
> **Author response**
>
> We thank the reviewer for their comments about our work.
>
> **[Applying our methodology to other domains]** We do agree that our methodology is geared towards ML tasks where we have access to metadata (e.g., labels or attributes for every data point) with an underlying class structure. However, we do not see this as particularly limiting as such tasks abound in several domains including images, text, speech, videos. We focus on images as, arguably, this is one of the canonical settings where new architectures and algorithmic tools are currently being developed.
>
> We will expand the discussion in Section 4.2 to elaborate on these points and also include examples of benchmarks from other domains where the BREEDS methodology could be readily applied. Example tasks include sentiment analysis (e.g., grouping by “metropolitan area” in [Yelp](https://www.yelp.com/dataset) or by “brand” in [Amazon reviews](http://jmcauley.ucsd.edu/data/amazon/)), visual question answering (e.g.,  grouping by "topic" in [Stanford Question Answering Dataset](https://rajpurkar.github.io/SQuAD-explorer/explore/v2.0/dev/)), object localization and tracking (e.g., grouping by “facial attributes” in [CelebA](http://mmlab.ie.cuhk.edu.hk/projects/CelebA.html) or by “GPS location” in [Berkeley DeepDrive](https://bdd-data.berkeley.edu/)).
>
> **[Formalization]** The pipeline to create BREEDS benchmarks given a particular dataset and its corresponding class hierarchy is as follows:
> 1. Choose a level in the hierarchy and use it to define a set of “superclasses” by grouping the corresponding dataset classes together. Note that the original dataset classes form the subpopulations of the superclasses.
> 2. For every superclass, select a (random) set of subpopulations (i.e., classes in the original dataset) and use them to train the model to distinguish between _superclasses_ (we call this the source domain).
> 3. For every superclass, use the remaining unseen subpopulations  (i.e., classes in the original dataset) to test how well the model can distinguish between the superclasses (target domain).
>
> We will include this description along with pseudocode in the appendix.

---

### Author Response · Authors · 2020-11-25
**Revised manuscript**

We thank all the reviewers for their comments and feedback.

We uploaded a revised version of the manuscript that contains the edits we mentioned in our responses. Specifically:
- In Section 4.2, we discuss how our methodology can be applied beyond classification tasks.
- In Appendix A.2, we provide pseudocode for our pipeline.
- In Section 6, we emphasize how our setup differs from the typical domain generalization setup.

We hope that these edits address the corresponding concerns raise by the reviewers.

---

### Decision · Program_Chairs · 2021-01-07
**Final Decision**

**Decision:**

Accept (Poster)

**Comment:**

I agree with the reviewers' positive comments about the paper. The BREEDS approach to generating benchmarks seems to be a useful one and addresses an important problem in the space. This approach could be the start of a nice direction of inquiry that will give us new insights into subpopulation shift. And most of the reviewers' negative concerns were addressed by the revision.